# Soil Lead (Pb) in New Orleans: A Spatiotemporal and Racial Analysis

**DOI:** 10.3390/ijerph18031314

**Published:** 2021-02-01

**Authors:** Sara Perl Egendorf, Howard W. Mielke, Jorge A. Castorena-Gonzalez, Eric T. Powell, Christopher R. Gonzales

**Affiliations:** 1Cornell Atkinson Center for Sustainability and School of Integrative Plant Science, Cornell University College of Agriculture and Life Sciences, Ithaca, NY 14853, USA; 2Department of Pharmacology, Tulane School of Medicine, New Orleans, LA 70112, USA; jcastorenagonzalez@tulane.edu (J.A.C.-G.); chrisgc99@gmail.com (C.R.G.); 3Lead Lab, Inc., New Orleans, LA 70119-3231, USA; powellet2@gmail.com

**Keywords:** community disparities, environmental justice, lead exposure, blood Pb, comorbidity

## Abstract

Spatialized racial injustices drive morbidity and mortality inequalities. While many factors contribute to environmental injustices, Pb is particularly insidious, and is associated with cardio-vascular, kidney, and immune dysfunctions and is a leading cause of premature death worldwide. Here, we present a revised analysis from the New Orleans dataset of soil lead (SPb) and children’s blood Pb (BPb), which was systematically assembled for 2000–2005 and 2011–2016. We show the spatial–temporal inequities in SPb, children’s BPb, racial composition, and household income in New Orleans. Comparing medians for the inner city with outlying areas, soil Pb is 7.5 or 9.3 times greater, children’s blood Pb is ~2 times higher, and household income is lower. Between 2000–2005 and 2011–2016, a BPb decline occurred. Long-standing environmental and socioeconomic Pb exposure injustices have positioned Black populations at extreme risk of adverse health consequences. Given the overlapping health outcomes of Pb exposure with co-morbidities for conditions such as COVID-19, we suggest that further investigation be conducted on Pb exposure and pandemic-related mortality rates, particularly among Black populations. Mapping and remediating invisible environmental Pb provides a path forward for preventing future populations from developing a myriad of Pb-related health issues.

## 1. Introduction

Lead (Pb) exposure is responsible for more than one million deaths worldwide each year, and 24.4 million disability-adjusted life years [1]. According to a population-based cohort study by Lanphear et al. (2018), Pb is responsible for approximately 412,000 deaths in the US per year [2], which is “comparable to tobacco smoke as a leading cause of mortality” [3]. However, this exposure is often mediated by social factors, particularly race [4]. The disproportionate exposure and environmental injustice of Black and low-income communities to environmental Pb has long been recognized [5,6,7].

Exposure to Pb adversely impacts all bodily systems and is particularly associated with organ impairments to the heart, kidney, and nervous systems [8]. Pb is widely distributed throughout the body and the mechanisms of toxicity are common to all cell types, impacting organ systems over a wide range of blood Pb (BPb) (≤5–>50 µg/dL). Prior to the 1960s, the blood Pb guideline for acceptable exposure was 60 µg/dL. The guidelines of “acceptable” exposure decreased numerous times as analytical measurements improved and clinical studies indicated that adverse health effects appeared at lower levels of Pb exposure [9]. By 2012, researchers realized that “there is no known safe level of lead exposure” [10].

Tremendous advances have been made in reducing exposure to Pb, particularly by banning leaded gasoline, paint, and solder over the past few decades. However, the legacies of leaded gasoline, peeling paint, waste incineration and industrial activities are stored in environmental reservoirs, particularly in soil. Soil studies conducted in urban environments indicate that some communities, based on city size and location within the city, are disproportionately Pb contaminated. Urban studies found similar soil Pb patterns in Baltimore, cities in Minnesota, Louisiana, and UK cities [11,12,13]. The racial associations and environmental injustices of SPb have also been shown in Santa Ana, California, where census tracts with median household income <USD 50,000 had five times higher soil Pb concentrations than high-income census tracts [14]. Reducing the amount of Pb in the environments of communities also reduces blood Pb [15].

This study investigates temporal and spatial changes in environmental lead (Pb), children’s Pb exposure, and the ways in which this exposure are associated with social formations such as racial categories and socioeconomic status across the entire city of New Orleans. The data for this study were analyzed from a unique database compiled to measure the concurrent declines of soil Pb (SPb) and children’s blood Pb (BPb) over an interval of ~15 years in New Orleans [15]. Here we analyze the SPb and BPb data by several variables, including distance from the city center, residential racial population, and household income over two time periods. We hypothesize that Black communities in New Orleans are disproportionately exposed to environmental Pb in soil, causing increased BPb burdens. As we will discuss, the impacts of Pb exposure strongly overlap with immune dysfunctions, and many of the pre-existing health conditions that increase the severity of COVID-19, highlighting the need for further study of this topic. Our aim is to recognize specific and compounding environmental injustices in the midst of the global coronavirus pandemic—issues that can be remedied, particularly for the most vulnerable populations.

## 2. Methods

### 2.1. Soil Lead Data

In New Orleans, two soil Pb surveys were conducted that collected soil samples by census tracts (in 1998–2001 and 2013–2017). The two surveys were collected by the same personnel using the same extraction preparation methods and equipment and the same Spectro Inductively Coupled Plasma (ICP) analytical instrument [15]. The samples were collected in the same census tract locations using the 1990 boundaries [16]. The protocol was implemented consistently in both surveys and required the collection of nineteen surficial (2–3 cm depth) soil samples, as explained in detail in the following references [15,17]. After analysis, the median SPb results for each census tract were calculated. The datasets generated and analyzed for the current study are available in the Appendix A (Appendix A).

### 2.2. Children’s Blood Pb Data

Children’s blood Pb data were provided by the Louisiana Healthy Homes and Childhood Lead Poisoning Prevention Program (LHHCLPPP). There may have been a change in the sensitivity of the instruments used in the 2000–2005 BPb survey compared with the 2011–2016 BPb survey. The outcome was that the apparent lowest level of detection found in 2000–2005 was 3 µg/dL, while the lowest level of detection found in 2011–2016 was 1 µg/dL (see Figure 1 and Figure 2). Children’s BPb data for both New Orleans and the United States decreased during this time. Census tracts were included only if there were 5 or more BPb datapoints. The census tracts that included both SPb and BPb (*n* = 274) were matched. The BPb dataset collected by LHHCLPPP between 2000 and 2005 consists of 54,695 independent BPb results. The second LHHCLPPP BPb dataset was collected from 2011 to 2016 and consists of 27,249 BPb results from the same 274 census tracts [15].

### 2.3. Interpolated Soil Pb and Blood Pb Data

The median SPb or BPb levels were assigned to the center of mass of the corresponding 1990 census tracts [18]. Using these point values, continuous raster surfaces for the study area were created for SPb or BPb by Ordinary Point Kriging interpolation using Surfer 17 [19]. The surfaces had the following dimensions: 1821 columns and 1176 rows, with a cell size of 20 × 20 m^2^. The geodetic datum used was the North American Datum 1983 (NAD83), and the map projection was Universal Transverse Mercator, Zone 15 North (UTM 15N). The resulting surfaces were saved in Environmental Systems Research Institute (ESRI)Grid format. The interpolated values of SPb or BPb were extracted from corresponding Grids and joined to the corresponding center of mass of 2000 or 2010 US Census polygons. This was done using the Extract Values to Points Spatial Analyst Tool in ArcGIS 10.2 [20].

### 2.4. Spatial–Temporal Analysis

The New Orleans Main Post Office (MPO), 701 Loyola Ave, New Orleans, LA 70113, was selected as the central location of the city. Precedence for choosing the Main Post Office as the central location was established in a Baltimore urban garden study [11], particularly since the postal service was designed to maximize mail delivery efficiency in each city [21]. Distance analysis was conducted using the center of mass of census tracts as the distance from the MPO, estimated by the ArcGIS Point Distance Analysis tool [18]. The census tracts were divided according to distance by deciles. The raster cell values were appended to the tabular data associated with the census tract center of mass shape files. The census tract data on racial composition and socioeconomic data for the two surveys are from Integrated Public Use Microdata Series (IPUMS) [22].

### 2.5. Statistical Analyses

Statistical analyses were conducted using data-dependent permutation procedure models, such that outliers were not removed, nor were transformations of the dataset conducted [23,24]. The Multi-Response Permutation Procedure (MRPP) was used to calculate the exact moments of the underlying permutation distribution of all possible arrangements of the observed data (Table 1). This procedure produces statistical comparisons with distance function-based permutation tests and calculates statistical significance between treatments [25]. To measure the clinical, practical, or substantive significance of these statistical tests, we used the chance-corrected measure of effect size that is appropriate for MRPP, Mielke’s R. This measure of effect size provides a percentage of within-group agreement above what would be expected by chance [26]. Kendall’s tau-b rank correlation evaluates the strength of the associations between the variables. In this study, Kendall’s tau-b was calculated to evaluate pairs of observations to determine the overall strength of association based on the concordance and discordance of the ranked pairs (see Appendix A in the Appendix A) [27].

## 3. Results

### Environmental Signaling Over Time and Distance

Table 1 shows the census tract results divided into NEAR census tracts and FAR census tracts according to their distance from the Main Post Office (MPO). Table 1 lists the population density, percent racial composition, environmental Pb, children’s blood Pb, and household income characteristics. The differences between each of these characteristics for NEAR and FAR groups were analyzed with Multi-Response Permutation Procedure (MRPP) statistics [23]. The medians for soil Pb, children’s blood Pb, racial composition, and household income show that compared with FAR communities, NEAR communities have 9.3 or 7.5 times higher median SPb (*p*-values < 10^−34^), with a substantive Mielke’s R effect size of 35% in the 2000 survey and 30% in the 2015 survey. Children tested have higher median BPb concentrations by a factor of ~2 (*p*-values < 10^−35^), with an R effect size of 41% in 2000 and 34% in 2015. Annual median household income is lower by ~USD 16,000 or ~USD 13,000 (*p*-values < 10^−6^), with an R effect size of 15% in 2000 and 4% in 2015. With increasing distance from the city center, the percentage of White residents increases, and the percentage of Black residents decreases (*p*-values < 10^−5^), with R effect sizes of ~13% in 2000 and ~4% in 2015. As indicated by these results, White populations and income increase as SPb and BPb decrease with distance from the city center.

Comparing the two temporal surveys, children’s median BPb levels in NEAR and FAR communities between 2000–2005 and 2011–2016 exhibit the following Mielke’s R effect size and *p*-values, respectively (not shown in Table 2): all survey results for the two time periods—37% and 4.3 × 10^−92^; for NEAR vs. NEAR results—46% and 3.5 × 10^−51^; and for FAR vs. FAR results—76% and 1.3 × 10^−61^. We assume that the BPb results are representative of the total populations living in the NEAR and FAR groups of Metropolitan New Orleans. There were profound reductions in children’s BPb between 2000–2005 and 2011–2016 in NEAR and FAR groups.

Table 2 shows the New Orleans census tract sector characteristics as a function of distance sectors during two time periods. The spaciotemporal characteristics of race, SPb, BPb, and household income, along with other characteristics, are given. The top panel of Table 2 shows that, in 2000–2005, majority Black communities predominantly inhabited distance sectors 1–5, where the median SPb was 505 mg/kg. Majority White communities, distance sectors 6–10, inhabited communities where the median SPb for the sectors was 47 mg/kg. The bottom panel shows the characteristics of the distance sectors in 2011–2015. Especially noteworthy is the downward shift in both SPb and BPb. In sectors 1–5, the median SPb declined to 215 mg/kg (from 505 mg/kg) and in sectors 6–10 the SPb declined to 32 mg/kg (from 47 mg/kg). Black people were still more likely to inhabit high-Pb areas, but some changes are noted with black residents living in higher proportions in outlying sectors. Especially noteworthy is the decrease in children’s BPb. In 2011–2015, children’s median BPb in distance sectors 1–5 was 2.2 µg/dL (down from 5.7 µg/dL in 2000–2005). The decline in BPb was concurrent with the decrease in SPb, as described previously [15].

Figure 1 illustrates the findings listed in Table 2 by summarizing the SPb and BPb results and dividing them into 10 sectors by distance from the center of New Orleans (location of the Main Post Office or MPO). In 2000–2005, the median BPb at distances less than 6 km peaked at nearly 7 µg/dL and declined at distances >6 km from MPO to 3 µg/dL. For 2011–2016, the median BPb peaked at 3 µg/dL and declined to 1 µg/dL at 6 km. Children’s median BPb in New Orleans remains elevated closest to the center of the city and decreases with the distance from the MPO. These reductions are worthy of celebration, but there is no safe level of BPb in children [28].

Figure 2 was derived from data in Table 2 and shows the decrease in children’s median BPb by racial composition as a function of distance from the center of the city, where predominantly Black people live. The data display disproportionately high SPb in the NEAR vs. FAR groups of the city (Table 1 and Table 2, and Figure 2), largely representing Black vs. White populations.

## 4. Discussion

### 4.1. Spatial–Temporal Distribution of Soil Pb and Blood Pb: Environmental Injustice for Black Populations

In the present study, metropolitan New Orleans is observed through spatial–temporal dimensions. As shown in Table 1, children’s median BPb differences between NEAR and FAR groups in New Orleans are strongly associated with median SPb concentrations. This study notes that the predominantly Black populations living in the NEAR areas in 2000–2005 presented with a median BPb of 5.6 µg/dL. The BPb of children living in NEAR communities have consistently been higher than children living in FAR communities. Children living in high SPb communities perpetuate patterns of environmental racism, disproportionately burdening low-income communities of color [29,30].

Limited studies have documented the racial associations particular to SPb [14,31]. However, similar geographic SPb patterns have been found elsewhere [11,12], and all major cities contain historical Pb emissions from gasoline [32]. The findings presented here are in concordance with the inequitable health outcomes of vulnerable populations, which have long been associated with racial segregation and poverty [33,34,35], including as a result of Pb exposure [5]. For example, in the National Health and Nutrition Examination Survey III, conducted from 1999 to 2004, non-Hispanic Black children were nearly three times more likely than White children to have BPb > 10 ug/dL [36]. On the national scale, atmospheric Pb has been shown to positively correlate with percentages of Black children at the county level, while being inversely correlated to percentages of White youth [37]. The story of Flint, Michigan, has drawn attention to policies and practices that render Black, Brown and low-income communities disproportionately vulnerable to toxicant exposure [38,39]. While Pb exposure from these other environmental media have been widely acknowledged, our data show that environmental Pb, in the form of SPb, should be recognized as contributing to the racial disparities of BPb exposure.

Measuring SPb provides insights into adverse racial and socio-economic effects from an invisible source of urban inequity. Soil Pb is likely contributing to racialized health inequities in many more cities than are currently recognized. Despite knowledge about Pb exposure in various forms, societal institutions have been “slow to recognize the racial ecology of lead poisoning as a major form of health inequality” [4]. Childhood Pb poisoning has been identified as a “signature disease of poverty” [40], as evidenced in Table 1, which may, in part, explain the inertia from governing bodies and legislative authorities to address Pb poisoning. The lack of attention paid to the issue of environmental Pb can also be attributed to federal policies and corporate interests that approved the commercial addition of Pb in various products, notably gasoline and paint, in the first place [41].

Concurrent decreases in SPb and BPb exist between the two survey periods [15]. These reductions in BPb and the overall use of Pb as a result of major policy changes are encouraging. However, as observed in Table 1 and Figure 2, SPb results remain high (>190 mg/kg) closest to the center of the city, where Black and lower-income residents predominantly live, and SPb is consistently lower (<47 mg/kg) where White and wealthier populations reside. Empirically, this study shows that only communities where SPb is less than 47 mg/kg have an adequate margin of safety to curtail excessive Pb exposure in children.

### 4.2. Potential Implications for COVID-19

As mentioned in the introduction, the health impacts of childhood Pb exposure may be lifelong, and worldwide Pb is a leading cause of mortality [1,2,8]. In the midst of the global coronavirus pandemic, we would be remiss not to call attention to the overlap between the outcomes of Pb exposure and the pre-existing conditions that have contributed to COVID-19 severity and mortality. While the data presented above document children’s BPb, there is increasing evidence suggesting that childhood Pb exposure is linked with adult health outcomes [42]. The first column of Table 3 lists the dominant pre-existing health conditions associated with severe COVID-19 symptoms [43]. Each of these pre-existing conditions listed arises from Pb exposure, and relevant explanations and references are provided in the subsequent columns. Early reports in Italy demonstrated that 99% of those who died from COVID-19 had other illnesses, with 75% having high blood pressure (hypertension), 35% diabetes, and 33% heart disease [44]. In an autopsy report of Black people who died as a result of COVID-19 in Louisiana, all had at least one comorbidity, the most common conditions being hypertension and type 2 diabetes [45]. As Table 3 shows, Pb exposure is associated with each of these conditions. While associations are not necessarily causative, they necessitate further investigation.

We call attention to these overlaps and question the extent to which people with morbidities from Pb exposure are more susceptible to becoming severely ill with COVID-19. Testing such a hypothesis would require measuring the bodily burden of Pb, but such data were not available due to Health Insurance Portability and Accountability Act (HIPAA) restrictions. Even if such data were safely and anonymously available, Pb is stored in bone tissue and while noninvasive bone Pb analyses are available, they are not frequently performed. Hence, cumulative Pb impacts are commonly underrecognized [8]. Understanding the linkages between Pb-associated morbidities and other adverse health issues such as COVID-19 would enable effective public health decisions to address and prevent exposure. Outside of general data on COVID-19, actual information on spatial–temporal infections with the new coronavirus, SARS-CoV-2, in the NEAR and FAR groups of New Orleans is lacking. Such data would also enable the verification of this hypothesis.

### 4.3. Actions for Primary Prevention for Soil Pb

The city-wide datasets for soil Pb, blood Pb, racial and socioeconomic factors, analyzed by distance from the city center and over time, indicate that disproportionate burdens of Pb exposure are placed on already vulnerable—and specifically Black and low-income—populations in New Orleans. Action is needed to effectively prevent exposure to an invisible source of environmental Pb before health issues arise, and primary exposure prevention to SPb is surprisingly feasible. Contaminated soil can be affordably covered and, as long as the cover is maintained, primary prevention can be achieved [58]. The Norwegian parliament set a precedent in 2006 and established a national clean soil action program [59]. The New York City (NYC) Mayor’s Office of Environmental Remediation established a Clean Soil Bank, which is the first municipal program in the U.S. to attempt such an endeavor [60]. The goals of the NYC program are to eliminate transporting clean excavation soil to landfills, save costs on soil imports, and enhance urban ecosystem services, as these soils mitigate SPb exposure. In so doing, the program may also address the systemic inequities that have been rendered undeniable by environmental justice organizing and research, as well as the data presented here. Lead is invisible and thus requires systematic mapping to understand the multiple ways in which exposure exacerbates urban environmental injustices. In particular, amidst the coronavirus pandemic, we call for further investigation into the adverse effects of Pb morbidity with severe COVID-19 outcomes. Soil lead cleanup should be considered, along with other measures, as a path forward to address an urgent and racially unjust environmental factor—one that will not only promote overall health but may also mitigate current and future pandemics.

## Figures and Tables

**Figure 1 ijerph-18-01314-f001:**
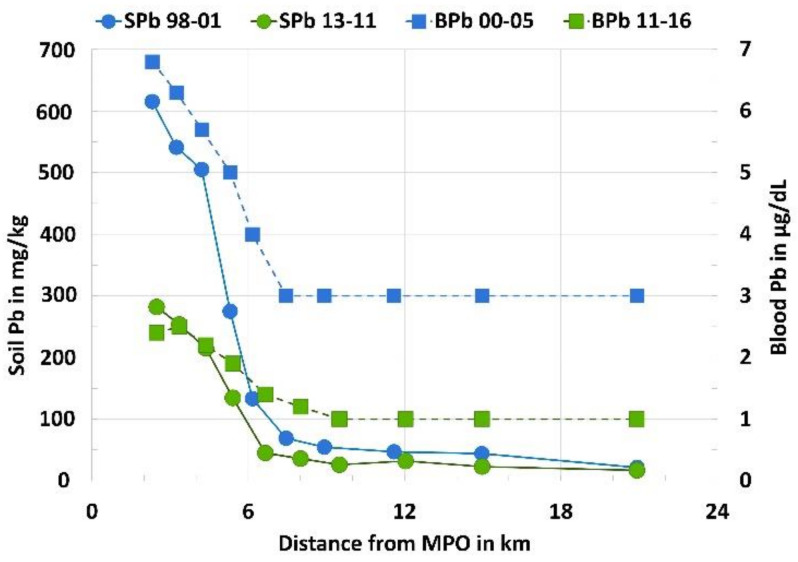
Soil Pb (circles with scale on the left) and blood Pb (squares with scale on the right) medians as a function of distance sectors from the Main Post Office (MPO) for the 1998–2001 soil Pb and 2000–2005 blood Pb surveys compared with the 2013–2017 soil Pb and 2011–2016 blood Pb surveys.

**Figure 2 ijerph-18-01314-f002:**
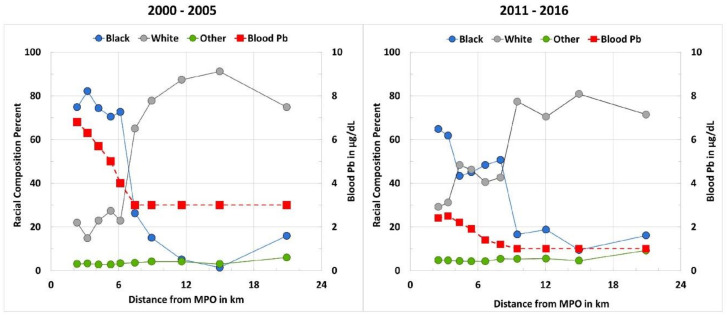
The relationship between racial composition and children’s median blood Pb as a function of distance sectors from the Main Post Office (MPO) to outlying communities. Children’s blood Pb (BPb) data (red squares with the scale on the right) are not disaggregated by race. The overall racial composition of the sampling areas are represented (blue, grey, and green circles with the scale on the left).

**Table 1 ijerph-18-01314-t001:** The percentage of White, Black, and Other populations, median interpolated soil Pb, median interpolated blood Pb, and median household incomes in 2000 and 2015 for NEAR and FAR groups from the main post office. The median household income data are from IPUMS [22].

	Dist. MPO km	Area km^2^	Pop.km^2^	% White	% Black	% Other	SPb mg/kg	BPb µg/dL	Household Income US$
2000									
NEAR									
N of Census Tracts	147	147	147	147	147	147	147	147	147
Minimum	0.0	0.1	294	0.0	0.9	0.3	35	3.0	4621
50%	3.8	0.6	3892	23.0	72.7	3.1	410	5.7	21,981
Maximum	6.2	6.4	15,527	97.4	99.5	24.8	1774	10.6	109,721
FAR									
N of Census Tracts	147	147	147	147	147	147	147	147	146
Minimum	6.2	0.5	22	0.0	0.0	0.0	6	2.1	16,250
50%	10.2	1.6	2388	78.1	12.7	4.2	44	3.0	37,919
Maximum	20.9	18.8	6476	98.7	100.0	21.5	237	5.0	146,158
*p*-Value	2.5 × 10^−53^	1.7 × 10^−26^	1.4 × 10^−19^	5.2 × 10^−14^	1.8 × 10^−14^	4.3 × 10^−5^	4.8 × 10^−40^	2.1 × 10^−42^	2.4 × 10^−22^
Mielke’s R	0.425	0.143	0.127	0.130	0.138	0.028	0.349	0.411	0.149
2015									
NEAR									
N of Census Tracts	143	143	143	143	143	143	143	143	141
Minimum	0.4	0.16	307	0.0	0.4	0.0	16	1.0	8738
50%	3.9	0.64	2813	40.8	51.5	4.4	187	2.1	30,917
Maximum	6.7	5.05	7702	97.0	100.0	21.1	910	4.9	155,714
FAR									
N of Census Tracts	143	143	143	143	143	143	143	143	143
Minimum	6.7	0.46	98	0.0	0.0	0.0	6	0.5	18,114
50%	10.6	1.44	2170	71.5	17.0	5.5	25	1.0	44,357
Maximum	20.9	18.65	6900	99.4	100.0	24.9	127	3.3	161,250
*p*-Value	4.2 × 10^−53^	8.7 × 10^−22^	1.2 × 10^−9^	5.8 × 10^−5^	2.0 × 10^−5^	0.016	1.4 × 10^−34^	2.6 × 10^−35^	5.1 × 10^−6^
Mielke’s R	0.437	0.115	0.062	0.036	0.042	0.010	0.301	0.335	0.035

**Table 2 ijerph-18-01314-t002:** New Orleans Census tract characteristics, soil Pb (SPb), and children’s blood Pb (BPb) are divided into 10 distance sectors from the Main Post Office (PO). The population density is inhabitants per square kilometer. The interpolated values for SPb, BPb, and household income are also given.

**SECTOR MEDIANS**										
**2000–2005**	**N**	**Main PO**	**Main PO**	**Pop.**				**SPb**	**BPb**	**Household**
**Sector**	**CTs**	**MIN DIST km**	**MAX DIST km**	**Density**	**% White**	**% Black**	**% Other**	**Soil Pb (interp.)**	**Blood Pb (interp.)**	**Income US$**
1	30	0.00	2.32	4506	21.90	74.80	3.08	615	6.8	11,950
2	29	2.36	3.25	5122	14.77	82.11	3.27	541	6.3	21,619
3	29	3.27	4.22	4129	22.99	74.37	2.81	505	5.7	21,456
4	30	4.30	5.31	3786	27.31	70.46	2.73	275	5.0	23,608
5	29	5.31	6.16	2589	22.85	72.60	3.37	133	4.0	27,526
6	29	6.19	7.45	2344	64.97	26.29	3.62	69	3.0	33,839
7	30	7.57	8.93	2377	77.75	15.04	4.13	55	3.0	40,014
8	29	8.97	11.60	2388	87.36	5.11	4.16	47	3.0	34,250
9	29	11.81	14.97	2577	91.18	1.38	3.03	44	3.0	40,086
10	30	14.97	20.92	1990	74.79	15.92	6.03	22	3.0	39,699
TOTAL	294									
**SECTOR MEDIANS**										
**2011–2015**	**N**	**Main PO**	**Main PO**	**Pop.**				**SPb**	**BPb**	**Household**
**Sector**	**CTs**	**MIN DIST km**	**MAX DIST km**	**Density**	**% White**	**% Black**	**% Other**	**Soil Pb (interp.)**	**Blood Pb (interp.)**	**Income US$**
1	29	0.38	2.49	3610	29.17	64.74	4.78	282	2.4	25,389
2	28	2.50	3.36	3632	31.16	61.83	4.67	254	2.5	28,927
3	29	3.38	4.38	2707	48.33	43.29	4.37	215	2.2	34,583
4	28	4.38	5.41	2582	46.18	45.07	4.30	135	1.9	34,822
5	29	5.43	6.66	1981	40.48	48.30	4.27	45	1.4	41,172
6	28	6.73	8.01	2076	42.63	50.68	5.39	36	1.2	38,312
7	29	8.05	9.50	1980	77.36	16.56	5.35	26	1.0	44,984
8	28	9.56	12.04	2216	70.40	18.77	5.49	32	1.0	39,224
9	29	12.24	14.98	2233	80.83	9.44	4.54	23	1.0	52,422
10	29	15.03	20.92	2040	71.46	16.07	9.22	17	1.0	47,679
TOTAL	286									

**Table 3 ijerph-18-01314-t003:** Pb exposure and COVID-19 comorbidity conditions.

Pre-Existing Condition for COVID-19 Comorbidity	Association with Pb Exposure	References
High blood pressure (Hypertension)	Most studied cardiovascular outcome of Pb exposure, effects may occur at BPb ≤ 5 µg/dL.	Apostoli et al., 1990 [46]; Bertin de Almeida Lopes et al., 2017 [47]
Coronary heart disease (CHD)	A positive dose–response relationship at BPb ≤ 10 µg/dL is an all-cause mortality and mortality cause of coronary heart disease.	Menke et al., 2006 [48]; Schober et al., 2006 [49]
Chronic obstructive pulmonary disease (COPD)	Over a BPb range of ≤10 µg/dL and BPb > 50 µg/dL, workers had decreased pulmonary function, obstructive pulmonary disease, increased asthma, and shortness of breath compared to controls.	Chung et al., 2015 [50]; Pugh Smith and Nriagu, 2011 [51]
Chronic kidney disease (CKD)	Exposure of BPb (≤ 10 ≥ 50 µg/dL) alters kidney function and chronic kidney disease (CKD). Renal nephrotoxicity severity is associated with increasing BPb.	Muntner et al., 2003 [52]; Pollack, 2015 [53]
Inflammation, immune system disorders and lymphatic system dysfunction	Elevated BPb (≥10 µg/dL) is an important factor in autoimmune diseases, chronic inflammation, and edema.	Boskabady et al., 2018 [54]; Mishra et al., 2009 [55]; CDC, 1991 [56]; Aghdam et al., 2019 [57]

## Data Availability

The complete dataset for this study is available from the IJERPH website at www.mdpi.com/xxx/s1.

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
