# Peer review of "Soil Lead (Pb) in New Orleans: A Spatiotemporal and Racial Analysis"

_ijerph, 2021, doi:10.3390/ijerph18031314_

Round 1

Reviewer 1 Report

A useful and potentially important study, but I have one major and a few less major comments that need to be addressed in a revision.

  1. major.

    Truth in advertising: The paper is presented as relating to covid-19. In fact there is nothing new presented about covid-19. The paper compares soil and childrens' blood lead concentrations in two New Orleans regions at two time points ten years apart. Interesting and important results but the authors do not have any substantive content related to covid-19.

    The authors speculate that Pb is a "contributing comorbidity factor for COVID-19". There are no data to support this conclusion, indeed no data about comorbidities for covid-19 at all. 

    The blood Pb data presented here are for children of unspecified ages. Table 1 summarizes previous study results relating blood Pb in adults with various chronic health conditions that are comorbidities for covid-19. But those chronic conditions occur in middle-aged or older adults. The connection between blood Pb in children and comorbities for covid-19 is very tenuous, not enough to justify claims made in the paper. 

    Please delete mention of covid-19 from the paper, it is a distraction for an otherwise meritorious study,

    2. less major

    a. Misuse of statistics.  abstract and paper give extremely low p values for comparisons of blood and soil Pb. A p value in null hypothesis significance testing is the probability that differences equal to or greater than those observed will be found assuming the null hypothesis (that the groups being compared are precisely the same and assuming no systematic error). But disproving the null hypothesis is not interesting since we already know that it is false. Also the statistical model assumed in NHST is not precise enough to allow reliable calculation of such tiny probabilities in any event. 

    p value is not a measure of effect size and comparison of (essentially meaningless) p values on p. 4 and elsewhere adds little. In the abstract and elsewhere please state the effect size directly and discuss its biological significance. 

    b. Please describe the results more completely, use appropriate whisker plots or histograms to show the distribution of values for blood and soil lead. I wonder whether the differences between the differences in BPb in the "near" and "far" groups in the most recent data set are large enough to be meaningful. The interest in this paper would be improved if you could identify a census tract with exceptionally high Pb exposure where further remediation efforts would pay off the most.

    c. This study appears to be an extension of Mielke's 2019 PNAS paper (cite 41) using the same data. If so please acknowledge. Have the BPb results been published before?

    d. Fig. 1 and elsewhere show a striking change at 6 km from the MPO, which cannot be a real effect. It is probably an artifact of analysis, perhaps from grouping data from different census tracts.  Please explain. Does a more correct analysis reduce or strengthen the differences between "near" vs. "far" populations?

    e. The writing has a polemical edge that detracts from an appearance of objectivity. Paper would be strengthened by toning it down. The one striking effect I see between the old and new data sets is the substantial decline in BPb in both "near" and "far" populations - is that not a cause for celebration?

Reviewer 2 Report

The authors in this manuscript describe an ecological study in which they examine the association between soil and blood lead levels, derived from two separate databases, with distance from the centre of New Orleans.  They then state that these associations help to determine the impact of COVID-19 in Black populations due to previously reported associations between blood Pb levels and co-morbidities associated with Covid-19. 

There are significant concerns regarding this paper.

  • it does not provide any linkage between the soil Pb and blood Pb levels with levels of morbidity or mortality in New Orleans. The provision of this data would be essential to support their hypothesis. At present, without such data, it is a simple assertion that Pb is important. In addition, reference 4, line 149 used to support the statement “the disparate COVID-19 severity of Black residents in New Orleans” does not appear to provide such data.  
  • the section entitled “Lead exposure as a pre-condition for Covid-19 comorbidity” in the Results section is not a result as it is simply a description of pre-existing conditions for Covid-19 comorbidity associated with PB exposure. This material should be either in the introduction or the discussion. However there is an additional issue in that the references again don’t necessarily provide support for the statements in Table 1. For example, the authors state that diabetes is associated with BPb levels of ≤10mg/dL and cite Moon et al (2013) and Tsiah et al (2004) as support for this statement. As far as this reviewer can judge neither paper supports this statement: indeed, the conclusion of the Moon paper is that “Blood lead, mercury and cadmium have no significant relationship with diabetes in the general Korean population.” This suggests that the material presented in Table 1 is based upon a very partial reading of the literature.
  • Figure 2 also provides some interesting data which does not necessarily agree with the stated hypothesis. For example, blood Pb levels in a predominantly White population at distances >6 km from the main post office in the time period 2000-2005 were all greater than the blood Pb levels in 2011-2016 whatever the distance from the main post office (and hence either predominantly Black or White populations). These would suggest that current blood lead levels are not that important a determinant of co-morbidity.

Other minor issues include

  • Lines 8, 10 the numbers 2 and 3 should be in superscript
  • Lines 12,13 etc All the reference numbers must be deleted
  • Line 14 “the co-morbidities” should surely be “some of the co-morbidities”
  • Line 21 BPb is not defined
  • There is no description of methods used for analysis of the soil and blood Pb and whether they changed during the different time periods
  • There is no detailed description of the statistical analysis carried out

Reviewer 3 Report

This is an interesting study on links between potential Pb exposure and covid19 comorbidities and one that I was thinking about myself last spring when things first started - so clearly there is interest in this idea.  Indeed, the authors findings suggest that further study is needed both to help better understand our current pandemic and importantly to better plan for the inevitable next one. 

Having said that, my enthusiasm for this manuscript at this time is somewhat tempered because of my confusion over how figure 2 is presented. It is possible I am not understanding the data, but, I think this suggests it needs to be better described in the results section and figure legend at a minimum. 

My specific confusion comes from the dual y axes. On the left is racial composition and on the right is blood lead levels in ug/dL. It appears that the median blood lead levels for the white population is actually HIGHER the further you go from the central city then the black population. This seems to be contradictory to everything we know about Pb and to the authors main argument. 

Again, I suspect that I am clearly confused about the data presentation - and as such urge the authors to display the data in a way that is easier to understand.

Reviewer 4 Report

Overall I thought the researchers did a nice job explaining the comorbidities between COVID and Lead. As stated in the discussion section this paper would have a much stronger argument and would be able to answer their hypothesis if they had been able to gain blood samples from COVID patients along with location of their homes. I believe that in the future this would be an appropriate study to run. As for this paper I wonder if the authors have been able to gain more data since submitting the article to better support their initial hypothesis.

Another analysis that might be nice in this paper would be if they could show/differentiate the age groups and exposure and also then differentiate from New Orleans COVID admittance, in general by age. I think that might be more insightful for how not only race but age plays a part in the comorbidities.

I believe if the authors could address some of these suggestions their paper will be more impactful.

Round 2

Reviewer 1 Report

revisions strengthened paper

modest final revisions requested:

  1. Remove the "p" values. The extremely tiny values (e.g. 10^(-35)) cannot possibly be meaningful, there is no way on earth that the population satisfies the assumptions in the statistical test with the extremely high degree of accuracy to allow numbers so far out in the tails of the distribution to have any meaning. Besides, the p value is a measure of consistency with the null hypothesis (precisely zero difference in populations and precisely zero systematic error). The null hypothesis is not interesting since you already know it is wrong. The meaningful quantity is the effect size and you now provide that.
  2. Some polemics still need to be removed: e.g. "render Black, Brown and low-income communities disposable"

Reviewer 2 Report

The major rewrite of this paper has improved it considerably. 

However the authors need to clarify whether the analytical methodology for Pb in soil or blood remained the same over the two periods and , if it had changed, how does this change affect the interpretation of the data.

I would also suggest that the first paragraph in the introduction (lines 30-37) is  in the wrong place and either should be at the end of the introduction or within the discussion

Author Response

Reviewer 2.

Dear Reviewer 2,

Thank you for your comments and suggestions, we have addressed each of them below.

  1. The major rewrite of this paper has improved it considerably.

Thank you for your suggestions for improvement.

  1. However the authors need to clarify whether the analytical methodology for Pb in soil or blood remained the same over the two periods and, if it had changed, how does this change affect the interpretation of the data.

We revised the methods to clearly state that the collection, preparation, and analysis of the samples were conducted by the same personnel using the same methods and analytical instrument. We refer to citations about our urban soil analytical methodology, and manuscripts are in print and available for anyone seeking to find out details about our techniques (see lines 215-219).

We added additional description about blood Pb collection to refer to the changes in analytical sensitivity and overall declines in blood Pb during the studied periods (lines 398-401).

  1. I would also suggest that the first paragraph in the introduction (lines 30-37) is in the wrong place and either should be at the end of the introduction or within the discussion

Thank you for this suggestion. We have combined the former first paragraph with the final paragraph of the introduction, now lines 200-211.

Thank you again for your time and assistance in improving this manuscript.